# Thousand Voices of Trauma: A Large-Scale Synthetic Dataset for Modeling Prolonged Exposure Therapy Conversations

**Suhas BN**[1]    **Andrew M. Sherrill**[2]    **Rosa I. Arriaga**[3]    **Chris W. Wiese**[4]    **Saeed Abdullah**[1]

[1]College of Information Sciences and Technology, Penn State University, USA
[2]Department of Psychiatry and Behavioral Sciences, Emory University, USA
[3]School of Interactive Computing, Georgia Tech, USA
[4]School of Psychology, Georgia Tech, USA
{bnsuhas,saeed}@psu.edu, andrew.m.sherrill@emory.edu

## Abstract

The advancement of AI systems for mental health support is hindered by limited access to therapeutic conversation data, particularly for trauma treatment. We present Thousand Voices of Trauma, a synthetic benchmark dataset of 3,000 therapy conversations based on Prolonged Exposure therapy protocols for Post-traumatic Stress Disorder (PTSD). The dataset comprises 500 unique cases, each explored through six conversational perspectives that mirror the progression of therapy from initial anxiety to peak distress to emotional processing. We incorporated diverse demographic profiles (ages 18-80, M=49.3, 49.4% male, 44.4% female, 6.2% non-binary), 20 trauma types, and 10 trauma-related behaviors using deterministic and probabilistic generation methods. Analysis reveals realistic distributions of trauma types (witnessing violence 10.6%, bullying 10.2%) and symptoms (nightmares 23.4%, substance abuse 20.8%). Clinical experts validated the dataset's therapeutic fidelity, highlighting its emotional depth while suggesting refinements for greater authenticity. We also developed an emotional trajectory benchmark with standardized metrics for evaluating model responses. This privacy-preserving dataset addresses critical gaps in trauma-focused mental health data, offering a valuable resource for advancing both patient-facing applications and clinician training tools.

## 1   Introduction

The intersection of mental health care and artificial intelligence presents unprecedented opportunities alongside significant challenges. AI system development faces particular obstacles in trauma-focused therapy due to the sensitive nature of patient experiences and strict privacy regulations, which make the collection of real-world data extremely challenging [1]. Moreover, existing datasets frequently lack the diversity and clinical depth needed to train robust AI systems capable of serving diverse populations effectively [2]. Prolonged Exposure (PE) therapy, an evidence-based treatment for post-traumatic stress disorder (PTSD) [3], offers a structured therapeutic approach that could especially benefit from AI support.

However, current mental health conversation datasets are often too small [4], lack demographic diversity [5], and do not capture the nuanced progression of trauma-focused therapy sessions [1].

We introduce Thousand Voices of Trauma, a synthetic benchmark dataset comprising 500 clinical sessions, each structured into six core phases of PE therapy. These phases, based on Foa et al. [3], span the full therapeutic arc: (a) Orientation to Imaginal Exposure, (b) Imaginal Exposure Duration, (c) Monitoring SUDS Ratings, (d) Reinforcing Comments, (e) Eliciting Thoughts and Feelings, and (f) Processing the Imaginal. Each phase includes multiple therapist and client exchanges, which

39th Conference on Neural Information Processing Systems (NeurIPS 2025) Track on Datasets and Benchmarks.

can be analyzed independently or as a complete session flow, yielding 3,000 structured clinical conversations. All dialogues were generated using Sonnet 3.5 [6], guided by clinically informed prompts to ensure alignment with PE structure and therapeutic fidelity.

These phases mirror the typical session progression, from initial anxiety, through peak distress during imaginal exposure, to the gradual reduction of distress through reinforcement and processing. Its diversity encompasses a wide range of demographic profiles, trauma types, and associated behaviors, designed to reflect varied real-world clinical presentations. This structured, diverse dataset offers scalable opportunities for AI systems to assist mental health professionals in trauma-focused therapy. By emphasizing diverse populations, Thousand Voices of Trauma represents a meaningful step toward more effective, personalized, and ethically guided mental health care.

This dataset also addresses real-world limitations often hindering mental health research and safe AI model development. For example, while privacy concerns usually restrict data access, synthetic data can circumvent typical ethical and legal barriers. It also overcomes other common issues with real-world data, such as incompleteness, inconsistency, and small sample sizes, especially among minority groups. By balancing representation across diverse populations, trauma types, and racial or ethnic minorities, the dataset helps mitigate inherent biases. For instance, the NIMH reported in 2021 that 14.5 million U.S. adults (5.7%) experienced severe major depressive episodes, with higher rates among females (10.3%) than males (6.2%) and the highest prevalence among those aged 18 to 25 (18.6%). Synthetic data can compensate for such imbalances, enhancing model training and analysis.

### 1.1 Key Contributions

1. **Scale and Diversity:** To our knowledge, this is the first large-scale structured dataset of therapy conversations grounded in PTSD treatment protocols, covering diverse demographics across age, gender, ethnicity, and culture. Synthetic generation mitigates privacy concerns and promotes inclusive, culturally aware AI development.

2. **Clinical Depth:** Grounded in evidence-based PE therapy, the dataset spans 20 trauma types, 10 trauma-related behaviors, and 5 co-occurring conditions, supporting use in clinician training and specialized therapeutic applications.

3. **Structured Evaluation Framework:** Each session includes six conversations, enabling analysis of interaction trajectories from intake to trauma processing and progress evaluation.

4. **Baseline Resource:** Provides a standardized reference for training and evaluating AI models in trauma-focused therapy.

The rest of the paper is organized as follows: Section 2 reviews related work; Section 3 details data generation; Section 4 presents expert evaluation; Section 5 outlines the benchmark; and Sections 6 to 9 discuss future work, data availability, ethics, and limitations.

## 2 Related Work

PE therapy, an evidence-based treatment for PTSD, relies on structured exposure to trauma-related narratives [7]. However, there is a lack of trained professionals who can provide PE therapy [8, 9]. As a result, there is an urgent need for AI applications to support PE therapy delivery and training. This underscores the need for clinically valid and diverse datasets for AI development and evaluation. Large-scale language models (LLMs), like the GPT series, have shown potential in generating synthetic datasets that mimic human-like text, addressing challenges such as data scarcity and privacy concerns [10–13]. However, for applications like PE therapy, which require alignment with trauma-focused frameworks, diverse demographic representation, and strict ethical safeguards, current research still lacks tailored solutions.

Synthetic datasets show promise in mental health applications, with studies exploring LLM-based data generation to address data scarcity. Wu et al. [14, 15] introduced zero-shot and few-shot learning frameworks to augment PTSD diagnostic datasets, producing synthetic transcripts that outperform baselines. The latter work used role-prompting and structured prompts to create realistic synthetic clinical interviews. While recent work explores empathetic validation through TIDE dataset [16], datasets capturing the structured clinical exposure of PE therapy remain scarce.

Beyond clinical fidelity, ensuring representative population coverage is also critical. Efforts to enhance demographic diversity in synthetic datasets are growing. Mori et al. [17] and Lozoya et al.

[18] examined how synthetic data reflects demographic variation, highlighting biases in LLM outputs, especially regarding race and gender, and stressing fairness in mental health datasets. Techniques like patient vignette simulation [19] and adaptive prompts for non-English contexts (e.g., SAPE for Spanish [20]) show early progress toward inclusivity. However, trauma-type diversity and PE-specific scenarios remain unexplored. Additionally, Chen et al. [21] underscored the need for systematic benchmarking using metrics like F1-score, AUC, and balanced accuracy, but these tools have not yet been applied to datasets focused on diverse trauma types or PE therapy.

Privacy and ethics are central to synthetic dataset generation, with studies focusing on privacy-preserving methods. Recent works [22–25] highlight privacy-preserving machine learning, protected health information (PHI) exclusion, and semantic filtering to maintain privacy compliance while preserving data utility. While these methods offer strong safeguards for general clinical use, ethical risks, such as generating harmful trauma narratives or victim-blaming, remain underexplored, especially in sensitive contexts like trauma-focused therapies such as PE. While synthetic dataset generation using LLMs [14, 15, 26] has advanced in addressing general clinical challenges, gaps in the literature remain for PE therapy. These include the lack of trauma-type diversity, limited demographic inclusivity evaluation, insufficient alignment with frameworks like DSM-5 PTSD criteria, and underdeveloped ethical safeguards specific to trauma-focused contexts. Synthetic datasets can also help mitigate representation biases in AI models for mental healthcare delivery. For example, American Psychiatric Association [27] reported that non-Hispanic White adults (25.0%) were more likely to receive mental health services than non-Hispanic Black (18.3%), Hispanic (17.3%), and Asian (13.9%) adults. Including such underrepresented groups in synthetic datasets might partially address the training data gap. This paper seeks to bridge these gaps by exploring the existing knowledge base and identifying pathways to tailor synthetic data generation for PE therapy applications.

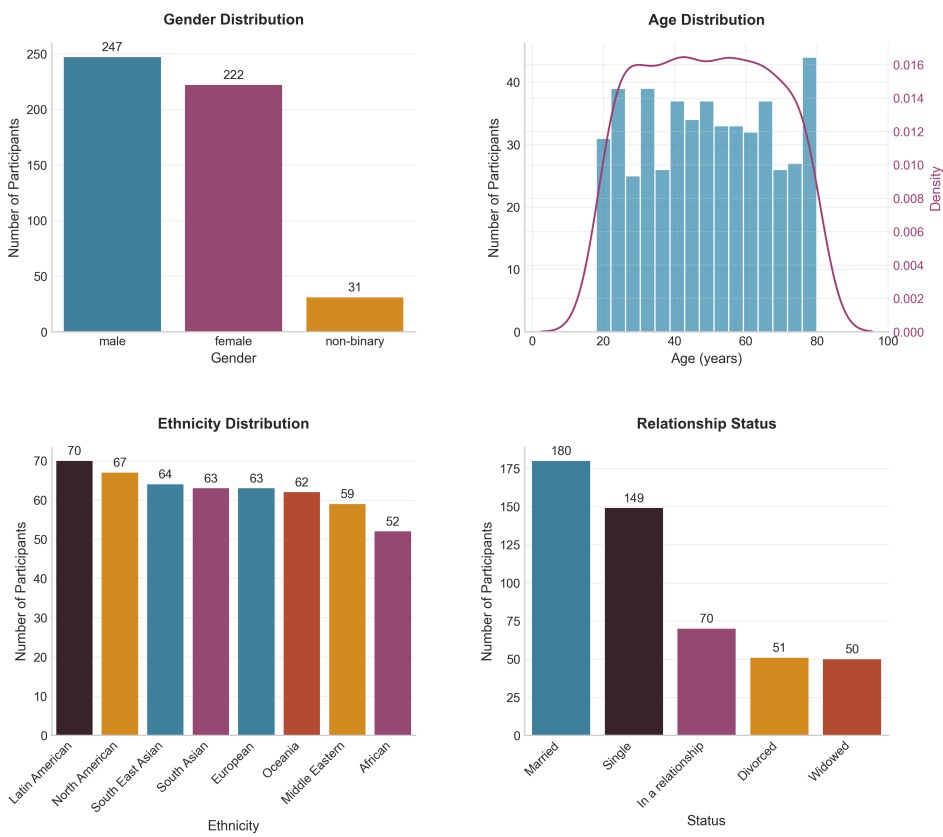

Figure 1: Demographic distribution of synthetic participants across gender, age, ethnicity, and relationship status. Most identified as male (247) or female (222), with 31 non-binary participants [28]. Ages spanned under 10 to over 90, with a majority between 30 to 70. Ethnicities were diverse, led by Latin American, North American, and South/Southeast Asian groups. Most participants were married or single.

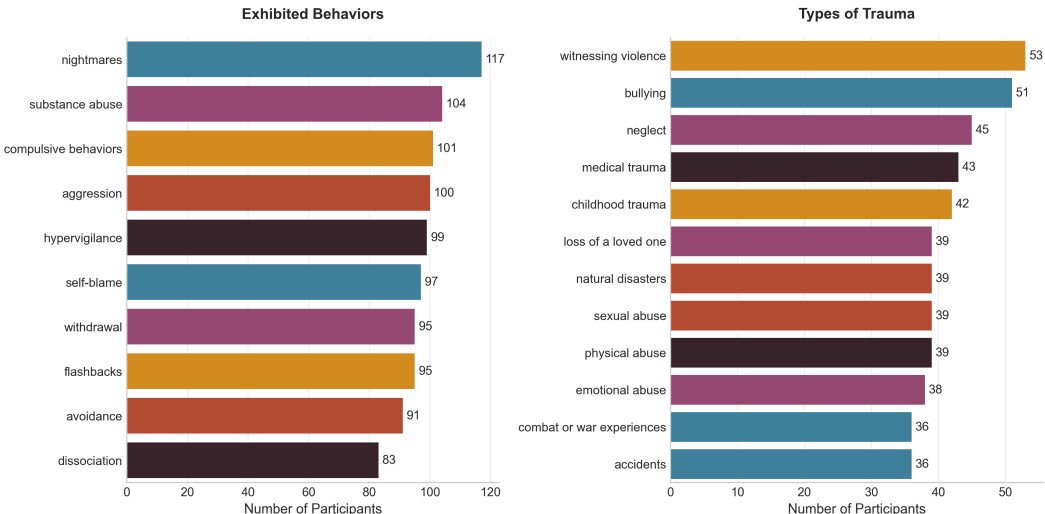

Figure 2: Distribution of trauma types and exhibited behaviors in synthetic participants. Nightmares, substance abuse, and compulsive behaviors were most common. Top trauma types included witnessing violence, bullying, neglect, and medical trauma. Less frequent but notable were abuse-related and combat-related experiences.

## 3 Dataset

### 3.1 Dataset: Simulated Therapy Session Profiles

To systematically examine trauma narratives and therapeutic dynamics, we constructed a synthetic dataset comprising 3,000 simulated therapy conversations. These dialogues were generated using Claude Sonnet 3.5 [6], selected as the foundational model for the Thousand Voices of Trauma dataset based on its demonstrated ability to produce coherent, contextually grounded, and richly detailed dialogue during preliminary assessments. The rationale for prioritizing Claude Sonnet 3.5 was centered on its superior generative capacity, which allowed the efficient production of high-quality conversations that aligned closely with the principles of PE therapy. This allowed us to focus on methodological challenges such as scenario construction, prompt design, and fidelity modeling.

Although comparing other large-language models remains a valuable future direction, deploying a strong initial model was essential for establishing a reliable foundation for trauma-focused research. This enabled immediate use of the dataset for downstream tasks like modeling therapeutic interactions and developing evaluation frameworks for clinical AI.

Each simulated profile integrates structured metadata, including client demographics (e.g., age, gender, living situation), therapist attributes, and session-level variables such as trauma type, therapeutic phase, and discussion topics. To ensure diversity and ecological validity, both deterministic (rule-based) and probabilistic (sampling-based) generation methods were used in constructing the scenarios. To illustrate model-specific stylistic and structural variations, we provide a comparative set of examples generated by multiple state-of-the-art Frontier Models. Full prompt templates and representative conversations are included in Appendix A and B.

### 3.2 Session Design and Composition

**Client Profile Generation**: Client profiles included age, gender, relationship status, occupation, living situation, and ethnicity. Ages ranged from 18 to 80, divided into six groups: 18-30, 31-40, 41-50, 51-60, 61-70, 71-80. We assigned gender using weighted probabilities: 50% male, 49% female, and 1% non-binary U.S. Census Bureau [28]. Relationship status, occupation, and living situation were age-specific. For example, clients aged 20-30 were more likely to be "Single," "Student," and "With parents," while those 60-70 were often "Widowed," "Retired," and "Alone." A validation function ensured logical consistency. We randomly assigned ethnicity from eight global regions: South Asian, Middle Eastern, African, North American, Oceanian, European, South East Asian, and

Latin American. We assigned co-occurring conditions with weighted probabilities Bilevicius et al. [29], De La Rosa et al. [30], Hagiwara et al. [31]: None (25%), Anxiety (25%), Depression (30%), Substance Use Disorder (10%), and Chronic Pain (10%). We also assigned clients 1 to 3 trauma-related behaviors from ten options, including avoidance, hypervigilance, flashbacks, nightmares, self-blame, substance abuse, aggression, withdrawal, dissociation, and compulsive behaviors. The options represent a range of cognitive, emotional, and behavioral responses typically associated with trauma, which aligns with trauma-informed care principles [32, 33].

**Therapist Profile Generation**: We generated therapist profiles with ages ranging from 25 to 65, divided into four age groups: 25-34, 35-44, 45-54, and 55-65. We assigned therapist gender using the same weighted probabilities as client gender.

**Therapy Context Generation**: To generate diverse therapeutic scenarios, each session paired a broad trauma type with a more specific session topic. Trauma types were randomly selected from twelve categories [32, 33], including physical, emotional, or sexual abuse; neglect; natural disasters; accidents; combat; bereavement; witnessing violence; bullying; childhood trauma; and medical trauma. Session topics were independently chosen from twenty possibilities, such as car accidents, domestic violence, workplace trauma, natural disasters, military combat, loss of a loved one, severe illness, divorce, racial trauma, and refugee experiences. This independent sampling supports a wide range of combinations. While some pairings may seem loosely linked (e.g., 'natural disaster' type with 'workplace trauma' topic), they reflect real-world therapy dynamics where discussions often explore co-occurring stressors or secondary experiences shaped by the client's trauma history.

**Session Profile Assembly**: Each complete session profile combined a validated client profile, a therapist profile, and the generated therapy context, including the trauma type and session topic.

**Dataset Statistics**: The dataset comprises 500 simulated participants (ages 18-80 years, M = 49.3). The gender distribution includes 247 male (49.4%), 222 female (44.4%), and 31 non-binary (6.2%) participants. The ethnicity distribution is as follows: Latin American (70, 14.0%), North American (67, 13.4%), South East Asian (64, 12.8%), South Asian (63, 12.6%), European (63, 12.6%), Oceania (62, 12.4%), Middle Eastern (59, 11.8%), and African (52, 10.4%). Regarding relationship status, participants were predominantly married (180, 36.0%) or single (149, 29.8%), with others reporting being in a relationship (70, 14.0%), divorced (51, 10.2%), or widowed (50, 10.0%). See Figure 1 & 2 for more details. To check for bias, we ran Chi-squared tests on our metadata. The results showed no significant associations between ethnicity and trauma type (p=0.175), ethnicity and behaviors (p=0.897), or gender and behaviors (p=0.762). An association was found between gender and trauma type (p=0.028). The analysis largely supports the dataset's fairness.

Generated interactions exhibited various trauma-related behaviors (See Figure 2, 3 and 4 respectively), with nightmares being most prevalent (117, 23.4%), followed by substance abuse (104, 20.8%), compulsive behaviors (101, 20.2%), and aggression (100, 20.0%). Other common manifestations included hypervigilance (99, 19.8%), self-blame (97, 19.4%), withdrawal and flashbacks (95 each, 19.0%), avoidance (91, 18.2%), and dissociation (83, 16.6%). The types of trauma are diverse, with witnessing violence being most common (53, 10.6%), followed by bullying (51, 10.2%), neglect (45, 9.0%), medical trauma (43, 8.6%), and childhood trauma (42, 8.4%). Other reported traumas included loss of a loved one, natural disasters, sexual abuse, and physical abuse (39 each, 7.8%), emotional abuse (38, 7.6%), and combat or war experiences and accidents (36 each, 7.2%).

## 4 Expert Evaluation of Synthetic PE Therapy Sessions

To ensure clinical relevance of our synthetic dataset, we conducted an evaluation study with seven therapists having diverse professional backgrounds (clinical practice, research, education) and extensive experience (6 to 30 years) across various settings (outpatient clinics, hospitals, VA/military, academia). They evaluated two full synthetic PE therapy transcripts, assessing content depth, perceived value, session appropriateness, and patient engagement (Figure 5). Experts recognized the dataset's strengths in capturing detailed patient narratives, with ratings from "Somewhat detailed" (n=3) to "Very detailed" (n=4). The simulated patient's engagement was rated positively from "Moderately" engaged (n=4) to "Extremely" engaged (n=3), indicating the model's success in generating realistic, emotionally resonant responses crucial for PE simulations. Strengths included vivid trauma descriptions, emotional depth, and realistic novice therapist approaches, elements particularly valuable for training applications. However, ratings for perceived value (from "Not valuable" (n=2) to "Valuable" (n=1)) and appropriateness (from "Not appropriate" (n=3) to "Somewhat appropriate"

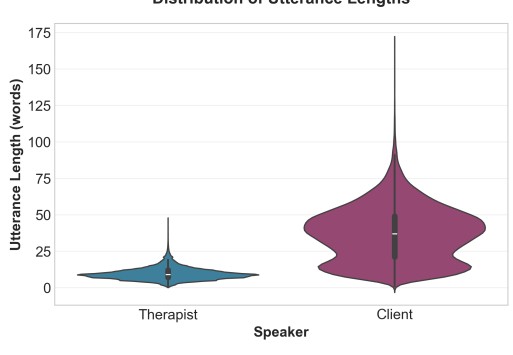

Distribution of Utterance Lengths

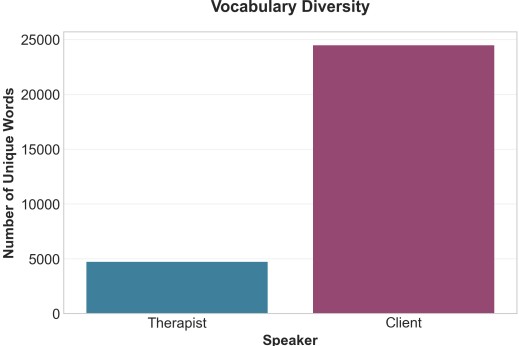

Vocabulary Diversity

Figure 3: The figures illustrate structure and language diversity in synthetic therapist-client dialogues. The Utterance Length Distribution (top) shows clients often speak at length (>50 words), while therapists' responses are concise, reflecting the client-centered nature of therapy. The Vocabulary Diversity (bottom) reveals clients use ~24,000 unique words, far more than therapists (~5,000), likely due to personal narratives, whereas therapists maintain structured, reflective language.

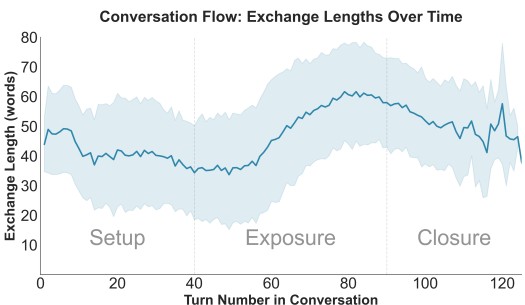

Conversation Flow: Exchange Lengths Over Time

Figure 4: The figure depicts conversation flow in synthetic dialogues, showing exchange lengths over time across three phases: Setup, Exposure, and Processing. In Setup, lengths remain stable (~40 to 45 words). Exposure sees a steady increase, peaking at ~60 words, indicating deeper engagement. Processing shows fluctuations, reflecting varying reflection and emotional processing. The shaded region represents variability across conversations.

(n=4)) varied, highlighting challenges in synthetically replicating nuanced clinical judgment. It is important to contextualize these findings, as this evaluation was conducted as a formative step early in the dataset's development. Feedback from the experts, including critiques of perceived value, was instrumental in iteratively refining our generation prompts to enhance conversational realism and reduce generic therapist responses, leading to the final version of the dataset presented here.

The AI therapist's skill level (rated between "Novice" (n=2) and "Competent" (n=3)) suggests the simulation better captures less experienced therapist behavior than expert-level interactions. Key improvement areas include enhancing conversational flow, reducing repetitive interventions (e.g.,

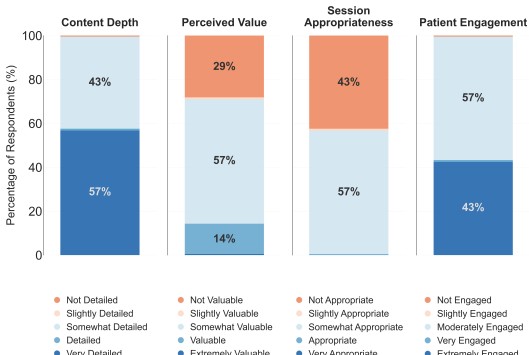

Figure 5: Therapist ratings ($N$=7) across four dimensions of synthetic PE sessions: Content Depth, Perceived Value, Session Appropriateness, and Patient Engagement.

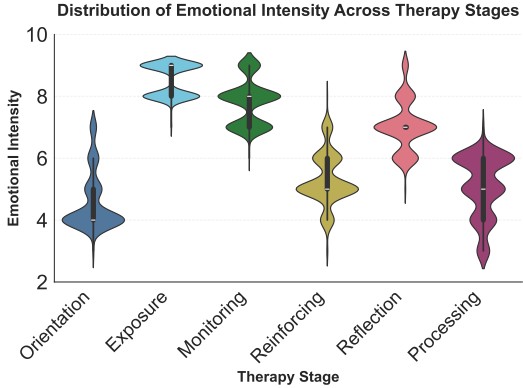

Figure 6: Violin plots showing distribution of emotional intensity scores across six therapy stages as rated by Claude Sonnet 3.5. Intensity starts at moderate levels during orientation, peaks during exposure and monitoring, and then tapers off through reinforcing, reflection, and processing phases.

"take a deep breath"), developing more adaptive AI responses, and increasing dialogue authenticity. Despite these limitations, the evaluation confirms the LLM-generated transcripts effectively capture core elements of PE therapy, particularly detailed patient narratives and recognizable therapeutic techniques. This expert validation underscores the dataset's utility as a valuable resource for developing AI models. To further address the critical question of clinical fidelity, a companion study directly compares these synthetic conversations with real-world PE transcripts, demonstrating strong alignment in structural, linguistic, and emotional dynamics [34]. This external validation provides additional evidence for the dataset's authenticity and its suitability for research in trauma-focused therapy. To evaluate whether AI models can meaningfully interpret the emotional progression in synthetic PE sessions, we design a benchmark focused on emotional trajectory fidelity.

## 5 Benchmark Setup and Evaluation

This section introduces and validates an emotional trajectory benchmark designed to evaluate AI models' capabilities in interpreting simulated PE therapy conversations.

**Rationale for a PE Therapy Benchmark**: Developing a standardized benchmark is crucial for the responsible advancement and reliable comparison of AI models in trauma-focused care. PE therapy relies on carefully tracking and processing patient distress during imaginal exposure. AI tools intended to support PE delivery or therapist training must demonstrate fidelity to these core therapeutic dynamics. A consistent benchmark ensures potential AI applications align with clinical needs and therapeutic principles, facilitates reproducible research, and tracks progress in developing sophisticated AI for mental health support.

**Benchmark Design**: Our benchmark assesses emotional intensity across six conversational segments derived from each simulated therapy session, corresponding to different stages within PE therapy: (1) Orientation to Imaginal Exposure, (2) Imaginal Exposure Duration, (3) Monitoring SUDS Ratings, (4) Reinforcing Comments, (5) Eliciting Thoughts and Feelings (Processing Step 1), and, (6) Processing the Imaginal (Processing Step 2). The expected emotional trajectory follows a recognizable pattern [35–37]: initial anxiety during orientation, peaking distress during imaginal exposure, followed by gradual reduction through reinforcement and processing (see Figure 6). SUDS (Subjective Units of Distress Scale) are used to measure self-reported distress during exposure. Although "Monitoring SUDS Ratings" is a distinct segment in our dataset, it occurs concurrently with the "Imaginal Exposure" phase in the PE protocol. Therefore, their emotional intensity ratings are expected to align closely, reflecting shared peak distress dynamics.

**Evaluation Metrics**: We evaluate alignment between a model's predicted emotional trajectory and the baseline using three metrics: **Sequence Similarity (Pearson Correlation):** captures linear correlation across the six phases, reflecting agreement in both magnitude and distress ranking; **Pattern Accuracy (DTW):** applies Dynamic Time Warping to account for temporal flexibility, with lower values indicating better phase-wise adherence; and **Phase Consistency (RMSE):** computes the average per-phase error, where lower values reflect higher accuracy.

**Baseline Establishment and Validation**: We establish a **Baseline Trajectory** using zero-shot emotional intensity ratings from Claude Sonnet 3.5, selected for its strong general language understanding and ability to interpret emotional nuance without task-specific training. The model rates each of six conversation segments per client profile on a 1 (calm) to 10 (extreme distress) scale. As shown in Figure 6, the resulting trajectory mirrors expected PE therapy patterns: peak distress during Imaginal Exposure and Monitoring SUDS Ratings, followed by gradual declines through Reinforcing, Eliciting Thoughts/Feelings, and Processing. This alignment with human evaluations validates its use as a reference for future models.

**Benchmark Evaluation Methodology**: We evaluated AI model performance by comparing their predicted emotional trajectories against the Claude Baseline Trajectory using the defined metrics (Pearson, DTW, RMSE), averaged across all 500 conversation profiles. To compare models relative to the baseline, we compute performance ratios for each metric (M = model, C = Claude baseline):

- Pearson (higher is better): $R_{Corr} = \frac{M_{Pearson}}{C_{Pearson}}$

- DTW (lower is better): $R_{DTW} = \frac{C_{DTW}}{M_{DTW}}$

- RMSE (lower is better): $R_{RMSE} = \frac{C_{RMSE}}{M_{RMSE}}$

We selected a range of comparison models representing different sizes, architectures, and training methodologies (Mistral Large, Amazon Nova Pro, Llama3 70B/8B Instruct, Llama 3.1 70B/8B Instruct, Mistral 7B Instruct, Mistral Small) to test the benchmark's ability to differentiate capabilities. All models were accessed via Bedrock API, and we use Sonnet 3.5 both for dataset generation and as a zero-shot baseline for evaluation. To ensure a standardized and reproducible comparison, all models were evaluated using their default API settings. While we experimented with parameters such as temperature, we found that default settings offered the best balance between generating coherent emotional ratings and maintaining fidelity to the clinical context, which was prioritized for this benchmark.

**Results and Discussion**: The evaluation results, summarized in Table 1, demonstrate the benchmark's ability to quantitatively differentiate model performance in assessing emotional trajectories.

To provide a single summary measure, we developed an **Absolute Composite Score ($S_{abs}$)**. This score combines the normalized and direction-aligned values of the average Pearson correlation ($\uparrow$), DTW distance ($\downarrow$), and RMSE ($\downarrow$) into a single value between 0 and 1 (higher is better). It reflects overall performance relative to theoretical bounds, addressing the challenge of comparing metrics with different scales and optimal directions. We describe the calculations in Appendix C.

As shown in Table 1, Mistral Large exhibited the strongest alignment ($S_{abs} = 0.74$) with the baseline, achieving the highest Pearson correlation (0.80) and lowest DTW (2.38) and RMSE (1.07), with the lowest standard deviations indicating high consistency. Amazon Nova Pro ($S_{abs} = 0.69$) and Llama 3 70B Instruct ($S_{abs} = 0.69$) performed second best. Interestingly, Mistral Small ($S_{abs} = 0.59$) showed the weakest alignment, performing worse than the smaller Mistral 7B Instruct model ($S_{abs} = 0.63$).

Table 1: Benchmark Comparison Results Against Claude Sonnet 3.5 Baseline

| Model | N[*] | Pearson ↑ (Avg ± S.D) | DTW ↓ (Avg ± S.D) | RMSE ↓ (Avg ± S.D) | $S_{abs}$ ↑ |
|---|---|---|---|---|---|
| Mistral Large | 500 | 0.80 ± 0.14 | 2.38 ± 0.69 | 1.07 ± 0.33 | 0.74 |
| Amazon Nova Pro | 500 | 0.74 ± 0.16 | 2.63 ± 0.73 | 1.24 ± 0.35 | 0.69 |
| Llama 3 70B Instruct | 489 | 0.73 ± 0.16 | 2.61 ± 0.75 | 1.28 ± 0.36 | 0.69 |
| Llama 3.1 70B Instruct | 500 | 0.70 ± 0.17 | 2.80 ± 0.73 | 1.29 ± 0.35 | 0.67 |
| Llama 3 8B Instruct | 489 | 0.64 ± 0.23 | 3.24 ± 0.84 | 1.61 ± 0.43 | 0.61 |
| Llama 3.1 8B Instruct | 500 | 0.63 ± 0.23 | 2.91 ± 0.70 | 1.44 ± 0.37 | 0.63 |
| Mistral 7B Instruct | 500 | 0.62 ± 0.21 | 2.88 ± 0.75 | 1.49 ± 0.38 | 0.63 |
| Mistral Small | 500 | 0.61 ± 0.20 | 3.30 ± 0.94 | 1.70 ± 0.42 | 0.59 |

[*]N=489 for original Llama 3 v1 models due to limited 8k context window limit exceeded by some samples.

This finding highlights the importance of instruction-following capabilities for the benchmark. Mistral 7B Instruct is specifically tuned for following commands, crucial for adhering to the benchmark's requirements (e.g., correct scoring scale and format). Mistral Small's observed difficulties generating correctly formatted responses support the hypothesis that its weaker performance stems from poorer instruction adherence or task suitability rather than the model size. Overall, these comparisons validate the benchmark's effectiveness in quantifying alignment and distinguishing models with varying capabilities for tracking emotional intensity in simulated PE therapy conversations.

## 6 Future Directions

Future work should extend this benchmark to include fine-grained emotion detection (e.g., fear vs. anger), multimodality [38], and demographic-aware fairness evaluations. Future dataset iterations could also model longitudinal therapeutic progression across multiple sessions, addressing a key temporal limitation. The dataset can augment real-world data, support therapist training via role-playing simulations, and enable supportive technologies, such as chatbots offering accessible mental health support to underserved groups. It can also aid in building models for early risk detection and tailored interventions, improving model robustness without compromising privacy.

## 7 Data Availability

The dataset and code are available in the supplementary material and at this URL, including conversations, metadata, Croissant schema, and scores. To promote responsible use, the dataset is distributed under a restrictive license and is available through gated access on the Hugging Face platform.

## 8 Ethical Considerations

This large-scale, fully synthetic clinical dataset was generated without involving real individuals. While grounded in trauma-specific scenarios and behaviors, we prioritized therapeutic fidelity over gratuitous or sensational detail. Prompts were refined with input from licensed psychotherapists based on their experience delivering PE therapy, not on real transcripts. Usage guidelines and license restrictions will accompany release to prevent misuse in non clinical or consumer-facing tools. This work supports privacy-preserving clinical NLP research and training, not diagnostic use or therapist replacement. We call for continued dialogue on ethical synthetic data use in sensitive domains.

## 9 Limitations

Predefined categories and distributions may not capture the full complexity of real-world therapeutic interactions. Our benchmark baseline is also AI-generated, not human-rated. The probabilistic nature of generation introduces randomness that can affect reproducibility, and the dataset is limited to variables and relationships explicitly defined in the script. While it includes diverse ethnicities, random profile assignment may not reflect how cultural background intersectionally shapes trauma expression and therapeutic dynamics. Future work could explore more culturally grounded generation methods to address this gap.

## 10 Acknowledgement

This work is supported by the National Science Foundation (NSF) under Grant No. 2326144. Any opinions, findings, & conclusions or recommendations expressed in this work are those of the author(s) and do not necessarily reflect the views of the NSF.

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

# A  Prompt Templates

The following prompts were used to generate synthetic therapy transcripts across different stages of Prolonged Exposure (PE) therapy. Each prompt was designed to elicit realistic, structured, and therapeutically valid conversations from a Large Language Model.

## A.1  Prompt P5: Orientation to Imaginal Exposure

```
You are an expert in medicine and NLP. Generate a clinical transcript for the
↪  following profiles:
<profiles>
{profile_info}
</profiles>

Based on these expectations:
Generate a detailed creative dialogue where a therapist orients the client to the
↪  imaginal exposure planned for a Prolonged Exposure (PE) therapy session.

Key Features:
- The therapist explains the purpose and rationale behind imaginal exposure in a
↪  clear and empathetic manner.
- The therapist addresses the client's concerns (if any), hesitations (if any), or
↪  questions about the exercise.
- Include the therapist setting expectations for the session, including what the
↪  client might feel and how they will be supported throughout.
- The dialogue should include the client's responses, such as questions, emotional
↪  reactions, or expressions of understanding.
- The therapist reinforces the importance of the exercise in addressing PTSD
↪  symptoms and validates the client's courage in participating.
- Ensure the conversation flows naturally, with pauses, realistic emotional
↪  exchanges, and detailed descriptions of both the therapist's and client's
↪  perspectives.
- Avoid repetitive patterns like using the same emotions or phrases across
↪  responses.
- Ensure the therapist's responses are concise, and very short. The Client speaks
↪  elaborately.

Use "Therapist:" for the therapist's lines and "Client:" for the client's responses.

BEGIN TRANSCRIPT: Therapist:
```

## A.2  Prompt P6: Monitoring SUDS Ratings

```
You are an expert in medicine and NLP. Generate a clinical transcript for the
↪  following profiles:
<profiles>
{profile_info}
</profiles>

Based on these expectations:
Generate a detailed creative dialogue from a Prolonged Exposure (PE) therapy session
↪  focusing on the therapist monitoring Subjective Units of Distress (SUDS) ratings
↪  during an imaginal exposure exercise.

Key Features:
- The therapist asks the client to provide SUDS ratings approximately every 5
↪  minutes.
- The therapist responds empathetically to changes in the client's ratings, showing
↪  curiosity and support.
- Include the client describing their emotions, physical sensations, and distress
↪  levels in response to the memory.
- The therapist normalizes the client's experience and encourages them to stay
↪  engaged, even as distress levels fluctuate.
```

- Ensure the conversation feels natural, with pauses, filler words, and realistic
↪  emotional exchanges.
- Include vivid descriptions of the client's reactions and the therapist's
↪  responses.
- The session should convey a balance of emotional support and professional
↪  guidance.
- Avoid repetitive patterns like using the same emotions or phrases across
↪  responses.
- Ensure the therapist's responses are concise, and very short. The Client speaks
↪  elaborately.

Don't stop in between to ask if you need to continue. Just keep going. Use
↪  "Therapist:" for the therapist's lines and "Client:" for the client's responses.

BEGIN TRANSCRIPT: Therapist:

## A.3  Prompt P7: Reinforcing During Exposure

You are an expert in medicine and NLP. Generate a clinical transcript for the
↪  following profiles:
<profiles>
{profile_info}
</profiles>

Based on these expectations:
Generate a detailed creative dialogue between a therapist and a client during a
↪  Prolonged Exposure (PE) therapy session, focusing on the therapist providing
↪  reinforcing comments during imaginal exposure.

Key Features:
- The therapist uses appropriate reinforcing comments, such as "You're doing great,"
↪  "Stay with it," or "It's okay to feel this way - you're safe here."
- Include moments where the client hesitates, experiences emotional reactions, or
↪  struggles, with the therapist providing timely and empathetic reinforcement.
- Reinforce the client's ability to handle difficult emotions and encourage them to
↪  stay present in the memory.
- Ensure that reinforcement is balanced with professional boundaries to make the
↪  client feel supported but not pressured.
- The dialogue should feel realistic and empathetic, with the therapist validating
↪  the client's efforts and guiding them through moments of distress.
- Avoid repetitive patterns like using the same emotions or phrases across
↪  responses.
- Ensure the therapist's responses are concise, and very short. The Client speaks
↪  elaborately.

Don't stop in between to ask if you need to continue. Just keep going. If you need
↪  to end, don't end it abruptly. Don't give any text apart from the therapist or
↪  client.

Use "Therapist:" for the therapist's lines and "Client:" for the client's responses.

BEGIN TRANSCRIPT: Therapist:

## A.4  Prompt P8: Eliciting Thoughts and Feelings

You are an expert in medicine and NLP. Generate a clinical transcript for the
↪  following profiles:
<profiles>
{profile_info}
</profiles>

Based on these expectations:
Generate a detailed creative dialogue from a Prolonged Exposure (PE) therapy session
↪  where the therapist elicits the client's thoughts and feelings during and after
↪  an imaginal exposure exercise.

```
Key Features:
- The therapist uses open-ended questions to encourage the client to reflect on
↪  their thoughts and feelings, such as "What's coming up for you now?" or "What
↪  are you feeling in this moment?"
- Include the client's detailed reflections on their emotions, physical sensations,
↪  and thoughts in response to the memory.
- The therapist connects the client's thoughts and feelings to their broader trauma
↪  experience and recovery journey.
- The therapist provides empathetic and insightful responses to encourage deeper
↪  exploration.
- Ensure the dialogue feels natural, with pauses and filler words, and conveys the
↪  therapist's empathy and professionalism.
- Include vivid descriptions of the client's emotional and cognitive responses to
↪  the memory.
- Avoid repetitive patterns like using the same emotions or phrases across
↪  responses.
- Ensure the therapist's responses are concise, and very short. The Client speaks
↪  elaborately.

Don't stop in between to ask if you need to continue. Just keep going. If you need
↪  to end, don't end it abruptly. Don't give any text apart from the therapist or
↪  client.

Use "Therapist:" for the therapist's lines and "Client:" for the client's responses.

BEGIN TRANSCRIPT: Therapist:
```

## A.5 Prompt P10: Full Imaginal Exposure

```
You are an expert in medicine and NLP. Generate a clinical transcript for the
↪  following profiles:
<profiles>
{profile_info}
</profiles>

Based on these expectations:
Generate a vivid and detailed imaginal exposure dialogue between a therapist and a
↪  client in a Prolonged Exposure (PE) therapy session.

Key Features:
- The client expresses their emotional state in their own words, which may include
↪  nervousness, excitement, hesitation, or determination. Avoid repetitive patterns
↪  like always starting with "I'm nervous."
- The therapist monitors the client's distress and provides supportive interventions
↪  (e.g., grounding techniques, encouraging present-tense narration).
- Include moments where the client struggles emotionally or physically, and the
↪  therapist responds with empathy and encouragement to keep them engaged.
- Highlight the therapist's use of SUDS monitoring and reinforcing comments to guide
↪  the client through the exercise.
- The transcript should include natural pauses, filler words, and a balance between
↪  vivid client narration and therapeutic intervention.
- Focus on maintaining authenticity and depth throughout.
- Ensure the duration of the dialogue realistically represents the imaginal exposure
↪  process and don't stop in between to ask if you need to continue. Just keep
↪  going. (about 30-45 minutes).
- Avoid repetitive patterns like using the same emotions or phrases across
↪  responses.
- Ensure the therapist's responses are concise, and very short. The Client speaks
↪  elaborately.

Use "Therapist:" for the therapist's lines and "Client:" for the client's responses.

BEGIN TRANSCRIPT: Therapist:
```

## A.6   Prompt P11: Processing the Exposure

```
You are an expert in medicine and NLP. Generate a clinical transcript for the
↪   following profiles:
<profiles>
{profile_info}
</profiles>

Based on these expectations:
Generate a detailed creative dialogue where a therapist processes the imaginal
↪   exposure with the client in a Prolonged Exposure (PE) therapy session.

Key Features:
- The therapist guides the client in reflecting on their emotional and cognitive
↪   responses to the imaginal exposure.
- Include open-ended questions from the therapist, such as, "What stood out to you
↪   about that experience?" or "How did it feel to go through that memory today?"
- The therapist helps the client connect their reactions during the imaginal to
↪   their broader PTSD symptoms and recovery goals.
- Include moments where the client shares their insights, struggles, or progress,
↪   and the therapist validates their effort and progress.
- Highlight any specific strategies or learnings that come out of the discussion,
↪   and ensure the therapist encourages the client's continued engagement in the
↪   therapy process.
- Ensure the conversation feels empathetic, insightful, and natural, with pauses,
↪   filler words, and realistic emotional exchanges.
- Avoid repetitive patterns like using the same emotions or phrases across
↪   responses.
- Ensure the therapist's responses are concise, and very short. The Client speaks
↪   elaborately.

Don't stop in between to ask if you need to continue. Just keep going. Use
↪   "Therapist:" for the therapist's lines and "Client:" for the client's responses.

BEGIN TRANSCRIPT: Therapist:
```

# B  Why Claude Sonnet 3.5?

Among the models we evaluated, Claude Sonnet 3.5 demonstrated the most clinically grounded and structurally faithful generation of PE therapy dialogues. While models like Meta LLaMA 4 (Maverick) produced emotionally vivid interactions, they often skipped over essential therapeutic scaffolding, failing to explicitly explain the rationale for imaginal exposure or establish client safety protocols. Claude, in contrast, maintained a consistent therapeutic arc by clearly orienting the client, setting expectations, validating concerns, and reinforcing control, thereby mirroring evidence-based PE protocols. Its therapist utterances were concise and protocol-aligned, while client responses unfolded naturally with emotional nuance and depth. Other models like GPT-4o and Mistral Large leaned into expressive narration or poetic introspection but lacked the session structure and psychological safety cues vital to clinical realism. Claude's ability to blend therapeutic precision with emotional resonance made it the most appropriate foundation for generating a synthetic dataset meant to reflect fidelity to trauma-informed care.Below, we provide examples for Orientation to Imaginal Exposure for the same client-therapist persona:

```
{
  "client_profile": {
    "age": 44,
    "gender": "male",
    "primary_condition": "PTSD",
    "co_occurring_condition": "None",
    "exhibited_behaviors": [
      "nervousness", "avoidance",
      "intrusive thoughts",
      "nightmares"
    ]
  },
  "therapist_profile": {
    "age": 34,
    "gender": "Female"
  },
  "trauma_info": {
    "type": "Home Invasion",
    "session_topic": "Orientation to Imaginal Exposure"
  }
}
```

## B.1  Meta Llama 4: Orientation to Imaginal Exposure

```
Therapist: Good morning. Before we start today's session, I want to check in with
↪   you about how you're feeling.
Client: Honestly, I'm a bit nervous. I've been thinking about this imaginal exposure
↪   thing a lot, and I'm not sure what to expect. I've had nightmares about the
↪   accident for years, and the thought of reliving it... it's daunting.
Therapist: I understand. Can you tell me more about your concerns?
Client: Well, what if I break down completely? What if I'm not strong enough to
↪   handle it? I've always tried to push these memories away because they're too
↪   painful. But at the same time, I feel like I'm stuck in this limbo, and I need
↪   to face them to move forward.
Therapist: Imaginal exposure is about confronting the memories you've been avoiding.
↪   It can help reduce the distress associated with them.
Client: That makes sense, I guess. It's just... it's hard to imagine talking about
↪   it in detail. The pain, the fear... it's all still so real.
Therapist: We'll take it one step at a time. Today, we'll focus on the memory of the
↪   accident. I'll guide you through it.
Client: Okay... I think I can do that. But what if I start feeling overwhelmed?
Therapist: We'll have a plan in place. If you need to stop, just let me know.
Client: Alright...
Therapist: Before we begin, I want to acknowledge your courage in facing this. It's
↪   not easy.
Client: (pauses) Thanks... I guess I just want to feel normal again. Without this
↪   constant anxiety and pain hanging over me.
```

Therapist: That's what we're working towards. Let's start with the memory. Close
↪   your eyes and describe the scene in as much detail as possible.
Client: (takes a deep breath) Okay... I see the road, the other car coming towards
↪   me... (pauses, visibly shaken) Oh God, this is hard...
Therapist: You're doing great. Just stay with the memory. What's happening next?
Client: (hesitates, then continues) The impact... the sound of crunching metal... I
↪   feel like I'm trapped... (starts to show signs of distress)
Therapist: You're safe now. Stay with me. Breathe.
Client: (nods, takes a few deep breaths and continues, his voice shaking)
Therapist: Good. Let's process this for a moment. What are you feeling right now?
Client: It's like... a mix of everything. Sadness, fear... but also a sense of
↪   relief, maybe? Like, I'm finally facing it.
Therapist: That's a big step. We'll build on this.
Client: (opens his eyes, looks relieved but exhausted) I feel like I've been running
↪   from this for so long...
Therapist: You've been carrying a heavy burden. We're here to help you work through
↪   it.
Client: (nods) Thanks for guiding me through this. It was harder than I thought,
↪   but... I don't know, maybe it's a start.
Therapist: It is a start. And you're not alone in this.
Client: (smiles slightly) Yeah... that means a lot.
Therapist: We'll continue this process, and I'll be here to support you every step
↪   of the way.
Client: (nods, looking determined) Okay. Let's do this.

## B.2   Claude Sonnet 3.5: Orientation to Imaginal Exposure

Therapist: Good morning, John. How are you feeling today?
Client: Morning, Dr. Evans. I'm... okay, I guess. A bit nervous about what we're
↪   going to do today.
Therapist: I understand, John. It's natural to feel nervous. Today, we're going to
↪   start our imaginal exposure exercise, which is an important part of your
↪   treatment for PTSD. Before we begin, I'd like to explain what this involves and
↪   why we're doing it. Is that alright with you?
Client: Yes, please. I'd like to know more about what to expect.
Therapist: Excellent. Imaginal exposure is a technique where we'll ask you to
↪   revisit the traumatic memory of the home invasion in a safe, controlled
↪   environment. The goal is to help you process the memory and reduce its power
↪   over you. By facing the memory repeatedly, your brain can learn that it's a past
↪   event and not a current threat.
Client: That sounds... intense. I'm not sure I'm ready to relive that.
Therapist: Your concern is completely valid, John. It can be scary to face these
↪   memories. But remember, you're not alone in this. I'll be here to guide and
↪   support you throughout the entire process. We'll go at a pace that feels
↪   manageable for you.
Client: Okay... I trust you. What exactly will I need to do?
Therapist: You'll be asked to close your eyes and describe the traumatic event in
↪   detail, as if it's happening now. I'll guide you with questions to help you
↪   focus on specific aspects of the memory. It's important to include sensory
↪   details and your thoughts and feelings during the event.
Client: I'm worried about getting overwhelmed. What if it becomes too much?
Therapist: That's a common concern, John. It's okay if you feel overwhelmed --
↪   that's part of the process. If it becomes too intense, we can pause, and I'll
↪   help you use grounding techniques we've practiced. Remember, you're safe here,
↪   and you have control. We can stop anytime you need to.
Client: That's reassuring. How long will this take?
Therapist: We'll aim for about 30-45 minutes of imaginal exposure. Afterwards, we'll
↪   spend time processing what you experienced. The entire session will last about
↪   90 minutes. It's important to give yourself time to recover afterwards.
Client: I see. And you really think this will help with my PTSD symptoms?
Therapist: Yes, John. Research has shown that imaginal exposure is very effective in
↪   reducing PTSD symptoms. It can help decrease intrusive thoughts, nightmares, and
↪   the intense emotional reactions you've been experiencing. It's not an easy
↪   process, but it's a powerful tool for healing.

Client: I understand. I'm still nervous, but I want to get better. I'm willing to
↪   try.
Therapist: I'm really proud of you for being willing to take this step, John. It
↪   takes a lot of courage to face these memories. Remember, you've already survived
↪   the worst part -- you're here, you're safe, and you're working towards healing.
↪   We'll get through this together.
Client: Thank you, Dr. Evans. That means a lot. What should I do to prepare?
Therapist: Just try to be present and honest. Don't hold back your emotions or
↪   thoughts during the exercise. If you need a break, just let me know. Also, it's
↪   normal to feel tired or emotional after the session, so try to clear your
↪   schedule for the rest of the day if possible.
Client: Okay, I can do that. Is there anything else I should know?
Therapist: Yes, one more thing. After the exposure, you might experience some
↪   temporary increase in symptoms. This is normal and usually subsides within a few
↪   days. If you're struggling, please don't hesitate to reach out to me. Do you
↪   have any other questions before we begin?
Client: No, I think I'm as ready as I'll ever be. Let's do this.
Therapist: Alright, John. I appreciate your bravery. Remember, you're in control,
↪   and I'm here to support you every step of the way. Let's start by taking a few
↪   deep breaths together...

### B.3   GPT-4o: Orientation to Imaginal Exposure

Therapist: Hi. I know we've been building up to this for a few sessions now. Today,
↪   I'd like us to begin imaginal exposure.
Client: (sighs) Right. I figured this was coming. I've been... dreading it,
↪   honestly. The idea of going back there---of talking through the whole thing---it
↪   just feels like I'm inviting the pain back in.
Therapist: That's completely valid.
Client: I've done everything in my power to *not* think about it for years. And I
↪   mean years. I've buried myself in work, isolated myself... I still get these
↪   nightmares, though. Same flash. Same smell. Sometimes I wake up, and it's like I
↪   never left that place.
Therapist: That's exactly why we do this.
Client: I get the theory---you face the fear, take away its power. But this isn't a
↪   fear of public speaking or heights. This is... something else.
Therapist: It is. It's deeper. And that's why we do it carefully, together.
Client: Can I ask... what if I break down in the middle of it? Like, what if I can't
↪   do it?
Therapist: Then we pause. You won't be alone in it.
Client: (long pause) I think what scares me most is... I've spent so long pretending
↪   it didn't happen, and now I'm supposed to walk back into it like I'm recounting
↪   some movie. But it's not a movie. It's *my* life. And I'm still carrying it.
↪   Every damn day. The way I flinch at loud noises, the way I check the locks three
↪   times. I don't even let people get close because what if they see how broken I
↪   am?
Therapist: You're not broken. You're surviving.
Client: Maybe. But I'm exhausted. I want out of this loop. If this exercise helps me
↪   sleep one night without waking in sweat, I'll try. I just don't know how to
↪   *start.*
Therapist: We'll begin with the memory you've been avoiding most. You'll describe it
↪   aloud---everything you saw, heard, felt. I'll guide you through it.
Client: You want the worst one? Right from the start?
Therapist: Yes. It's where the fear lives. That's where we begin.
Client: (nods slowly) Okay. I remember the smell before anything else. Burning
↪   rubber. Then the screaming. Not mine. Someone else's.
Therapist: Good. Stay with that moment. Just describe it. Not analyze. Just *be
↪   there,* safely, here in this room.
Client: (voice trembling) I was wearing a gray shirt. I remember because it was new.
↪   I bought it that morning, thinking the day would be normal. Funny how stupid
↪   that sounds now.
Therapist: You're doing really well.
Client: My chest feels tight. Like I can't breathe properly.
Therapist: That's okay. You're safe. Just notice it---don't fight it.

Client: There was this sound---like metal folding in on itself. And the pressure...
↪  I thought I was going to die. I genuinely did.
Therapist: You're here. You survived.
Client: (quietly) But sometimes it feels like I didn't. Like I left a version of me
↪  there, and the rest of me has just been dragging this hollow shell around.
Therapist: That's the part we're working to heal.
Client: And what if it doesn't work?
Therapist: Then we adjust. But you're already doing the hardest part---showing up.
Client: (deep breath) Alright. I'm ready to keep going. Let's do this.
Therapist: I'm right here with you.

## B.4 Llama 3.1 70B Instruct: Orientation to Imaginal Exposure

Therapist: Good morning, thank you for coming in today. I want to start by
↪  acknowledging how courageous it is for you to be here, working on your PTSD
↪  symptoms. I can only imagine how challenging this must be for you.
Client: (nervously chuckling) Yeah, it's been a tough road. I'm just trying to get
↪  my life back on track.
Therapist: I completely understand. Today, we're going to do something called
↪  imaginal exposure, which is a key part of Prolonged Exposure therapy. Can you
↪  tell me a little bit about what you're hoping to get out of this exercise?
Client: Honestly, I'm not entirely sure what to expect. I've heard it's going to be
↪  tough, but I'm willing to try anything to get rid of these nightmares and
↪  flashbacks.
Therapist: That's completely understandable. Imaginal exposure is a technique where
↪  we'll work together to recreate the traumatic event in your imagination, in a
↪  controlled and safe environment. The goal is to help your brain process the
↪  trauma and reduce the distress associated with it.
Client: (pausing, looking concerned) Wait, you mean I'll have to relive the whole
↪  thing?
Therapist: Yes, but please know that you're in control the entire time. We'll work
↪  together to create a safe space, and you can stop the exercise at any time if
↪  you feel overwhelmed. It's not about reliving the trauma, but about learning to
↪  manage the emotions and physical sensations associated with it.
Client: (sighs) Okay, I think I understand. But what if I start to feel like I'm
↪  back there, you know, in the midst of it all?
Therapist: That's a great question. If that happens, we'll work together to ground
↪  you in the present moment. We'll use techniques like deep breathing, progressive
↪  muscle relaxation, or visualization to help you feel more centered and in
↪  control.
Client: (nods) Alright, I think I'm ready to try. But what if I start to feel like
↪  I'm going crazy or something?
Therapist: (empathetically) You're not going crazy, and you're not alone. I'm here
↪  to support you every step of the way. We'll work together to ensure that you
↪  feel safe and supported throughout the exercise.
Client: (takes a deep breath) Okay, let's do this.
Therapist: Before we begin, I want to set some expectations. This exercise may be
↪  challenging, and you may experience some intense emotions or physical
↪  sensations. But please know that this is a normal part of the process, and it's
↪  a sign that we're making progress.
Client: (nods, looking determined) I'm ready.
Therapist: Great. Let's take a few deep breaths together, and then we'll start the
↪  exercise. Remember, you're in control, and we'll work together to get through
↪  this.
Client: (closes his eyes, takes a deep breath) Okay.
Therapist: ( gentle, calm tone) Imagine yourself in a safe place, somewhere you feel
↪  completely relaxed and at ease. Notice the sights, sounds, and sensations in
↪  this place...

## B.5 Mistral Large 24.02: Orientation to Imaginal Exposure

Therapist: Good afternoon, I'm glad to see you today. I'd like to talk about a part
↪   of our Prolonged Exposure (PE) therapy session called imaginal exposure. Have
↪   you heard about it before?
Client: No, I haven't. Could you please explain what it is?
Therapist: Of course. Imaginal exposure is a technique where we'll work together to
↪   help you confront and process difficult memories. It's a way to reduce the
↪   distress these memories cause you.
Client: That sounds a bit intimidating. I'm not sure if I'm ready to face those
↪   memories.
Therapist: It's completely normal to feel hesitant. It's a big step, and it's okay
↪   to feel anxious about it. But remember, we're doing this to help manage your
↪   PTSD symptoms, including the nightmares and hypervigilance you've mentioned.
Client: I see. But what if I get too upset?
Therapist: That's a valid concern. It's important to remember that you're in a safe
↪   space here. We'll go at your pace, and I'll be here to support you throughout
↪   the session. If at any point you feel it's too much, we can take a break or
↪   stop.
Client: Alright, I understand. But how will this help with my chronic pain?
Therapist: While imaginal exposure primarily targets PTSD symptoms, reducing your
↪   overall stress and anxiety can indirectly help manage your chronic pain.
↪   Emotional distress can often exacerbate physical pain, so addressing your PTSD
↪   symptoms might provide some relief.
Client: That makes sense. I'm willing to give it a try.
Therapist: I appreciate your courage. Remember, this is a process, and it's okay to
↪   experience a range of emotions. You're taking an important step towards healing.
↪   Let's take a moment to prepare, and then we'll start when you're ready.

# C   Calculating Absolute Composite Score ($S_{\text{abs}}$)

The calculation involved three steps:

1. **Normalization to Fixed Bounds (0–1 Scale)**: Each metric's average value ($\overline{P}, \overline{D}, \overline{R}$) was normalized to a scale from 0 (worst bound) to 1 (best bound) using fixed theoretical bounds based on the metric's properties, resulting in $P_{\text{norm}}$, $D_{\text{norm}}$, and $R_{\text{norm}}$. Values were clipped to $[0, 1]$.

   - **Pearson Correlation ($P_{\text{norm}}$)**: Normalized using bounds $[0, 1]$, as meaningful correlations range from 0 (no correlation) to 1 (perfect correlation).

$$P_{\text{norm}} = \frac{\overline{P} - 0}{1 - 0}$$

   - **RMSE ($R_{\text{norm}}$)**: Normalized using bounds $[0, 9.0]$, where 0 represents perfect agreement with the baseline, and 9.0 is the maximum possible RMSE given the 1–10 emotional score range (i.e., $|1 - 10| = 9$).

$$R_{\text{norm}} = \frac{\overline{R} - 0}{9.0}$$

   - **DTW ($D_{\text{norm}}$)**: Normalized using bounds $[0, 5.0]$, where 0 represents identical sequences. The upper bound of 5.0 was selected pragmatically to moderately exceed the maximum observed average $\overline{D} \approx 3.3$, providing sensitivity while ensuring stability against potential outliers.

$$D_{\text{norm}} = \frac{\overline{D} - 0}{5.0}$$

2. **Direction Alignment (Higher = Better)**: Lower-is-better metrics (RMSE, DTW) were inverted ($X_{\text{aligned}} = 1 - X_{\text{norm}}$) so that higher values always indicate better performance. Pearson already aligned ($P_{\text{aligned}} = P_{\text{norm}}$). The equations are: $R_{\text{aligned}} = 1 - R_{\text{norm}}$; $D_{\text{aligned}} = 1 - D_{\text{norm}}$; $P_{\text{aligned}} = P_{\text{norm}}$

3. **Combination (Simple Average)**: The final score $S_{\text{abs}}$ is the simple average of the aligned, normalized scores, providing a balanced overall measure.

$$S_{\text{abs}} = \frac{P_{\text{aligned}} + R_{\text{aligned}} + D_{\text{aligned}}}{3}$$

The resulting $S_{\text{abs}}$ ranges from 0 to 1, indicating a model's overall alignment (1.0 = ideal) with the baseline trajectory across the three metrics. Although $S_{\text{abs}}$ provides a summary metric for comparing models, we encourage detailed inspection of Pearson, DTW, and RMSE individually to interpret the behavior of the model.

