# OpenReview forum: "Thousand Voices of Trauma: A Large-Scale Synthetic Dataset for Modeling Prolonged Exposure Therapy Conversations"
_NeurIPS.cc/2025/Datasets_and_Benchmarks_Track — NeurIPS 2025 Datasets and Benchmarks Track spotlight_

### Official Review · Reviewer_y3Hw · 2025-06-27

**Rating:** 6
**Confidence:** 4

**Summary:**

This effort creates a synthetic dataset that can support the development of AI applications to support PE therapy delivery and training. Using the created synthetic dataset, the authors then introduce an emotional trajectory benchmark that they use to evaluate AI models.

**Additional Feedback:**

The paper was clearly written and well structured.

**Dataset Code Accessibility:**

Yes

**Dataset Code Comments:**

I was able to access and review the dataset.

**Ethical Considerations:**

No, there are no or only very minor ethics concerns

**Final Justification:**

I have reviewed the comments and am satisfied with my review.

**Limitations Weaknesses:**

When testing the benchmark across models, it is not clear how they managed the different models' settings.  These settings could have a significant impact on benchmark results in Table 1.  For example, is the temperature for all models set to zero, reducing the chances of aberrant model responses and reducing the impact of randomness from the probabilistic nature of LMMs.

**Strengths Contributions:**

This is a well-structured effort that provides both a valuable dataset and a benchmark.
1. The paper provides a convincing rationale for the need for AI datasets to support the development of AI applications to support PE therapy delivery and training. Additionally, they explain why synthetic datasets are preferable to real data (privacy concerns).  They produce analytics Figures 1 and 2 to demonstrate alignment of data with the diversity of patients. The authors also perform quality controls on the synthetic therapy dialog by having experts review the dialogs (section 4) and by applying logic constraints to the characteristics of the dialogs.
2. Section 5 presented the emotional trajectory benchmark. It provided explanations of how the benchmark was calculated.
3. Error analysis was included in the benchmark results (Table 1)

---

> ### Author Rebuttal · Authors · 2025-07-30
>
> Many thanks for your support and positive feedback on our paper. We’re glad the value of the dataset and the novel emotional trajectory benchmark came through so clearly.
>
> To clarify your question about model evaluation settings: all model evaluations used the default API settings provided by Amazon Bedrock. We chose this approach to ensure a standardized and reproducible comparison across models. While we did experiment with increasing the temperature setting to foster more “creative” or varied responses, we observed a clear trade-off: higher creativity often came at the cost of the model's adherence to core PE protocol fidelity. Given the clinical nature of the task, we therefore prioritized clinical alignment for the experiments presented in the paper. We will add this rationale to the final version.
>
> Thank you again for your enthusiastic review and insightful question.

---

> > ### Comment · Reviewer_y3Hw · 2025-08-05
> >
> > Thank your for your response

---

### Official Review · Reviewer_M2aq · 2025-06-29

**Rating:** 5
**Confidence:** 4

**Summary:**

This paper presents "Thousand Voices of Trauma," a synthetic dataset of 3,000 therapy conversations grounded in Prolonged Exposure (PE) therapy protocols for PTSD treatment. Using Claude Sonnet 3.5, the authors generated 500 unique cases across diverse demographics (ages 18-80, balanced gender representation, 8 ethnic groups) and 20 trauma types. Each case includes six conversational segments reflecting PE therapy progression from initial anxiety to emotional processing. The work consists of clinical validation by seven experienced therapists and establishes an emotional trajectory benchmark with standardised evaluation metrics.

**Dataset Code Accessibility:**

Yes

**Dataset Code Comments:**

The dataset meets high standards for accessibility and reproducibility. Comprehensive documentation, including prompt templates (Appendix A), model selection rationale (Appendix B), and methodological details, supports full transparency. The release of conversations, metadata, Croissant schema, and evaluation scores (Section 7) further ensures reproducibility.

**Ethical Comments:**

This work exemplifies strong ethical standards in sensitive clinical research. The fully synthetic approach eliminates privacy concerns while preserving clinical utility, directly addressing HIPAA and consent limitations. Involving licensed psychotherapists in prompt design ensures therapeutic fidelity without relying on real patient data. The dataset encompasses diverse demographics and realistic distributions of trauma types, promoting representation without sensationalism. Clear usage guidelines and licensing restrictions prevent misuse, and the resource is appropriately positioned for research and training, not diagnosis or therapy. Transparent discussions of limitations (Section 9) and validation outcomes (Figure 5) reflect intellectual honesty and a responsible approach to synthetic clinical data.

**Ethical Considerations:**

No, there are no or only very minor ethics concerns

**Final Justification:**

The authors provided a thoughtful and reasonable response to my concerns about the limited scale and cultural depth of the clinical validation. While the scope of therapist evaluation is narrow, their explanation regarding resource constraints and dataset release is acceptable. The paper’s core contributions remain strong, and I view the ethical safeguards and qualitative depth as appropriate for a first release. I am maintaining my score of 5.

**Limitations Weaknesses:**

- Validation Scale Limitations: While the clinical evaluation is thoughtfully designed, the involvement of only seven therapists constitutes a relatively small sample for establishing broader clinical consensus. Additionally, the evaluation focuses on just two full transcripts, which may not adequately represent the diversity of trauma types and demographic combinations in the dataset.

- Cultural Depth Considerations: Although the dataset includes eight ethnic groups, the paper does not explore how cultural factors might affect trauma expression, therapeutic rapport, or treatment approaches. The random assignment of ethnicity may overlook culturally specific trauma responses or clinical considerations that are critical in real-world settings.

- Temporal Scope Constraints: The dataset effectively captures single-session dynamics, but it does not model therapeutic progression over time. This limits its applicability for studying treatment trajectories, session-to-session evolution, or the development of longitudinal therapeutic relationships.

**Strengths Contributions:**

- Methodological Rigour and Clinical Grounding: The dataset is systematically constructed based on evidence-based PE therapy protocols, with clearly defined session phases that reflect real therapeutic progression.

- Comprehensive Validation Framework: Clinical evaluation by seven therapists with 6-30 years of experience across diverse settings (Section 4) demonstrates commitment to therapeutic fidelity. The multi-dimensional assessment (content depth, perceived value, session appropriateness, patient engagement) provides robust validation (Figure 5).

- Significant Scale and Diversity: The dataset addresses representation gaps with 3,000 conversations covering 20 trauma types, diverse demographics including age-appropriate relationship status and occupation assignments, and 10 trauma-related behaviours.

- Ethical Innovation: The use of fully synthetic data eliminates privacy concerns while preserving clinical realism and utility

- Reproducibility: The inclusion of detailed prompt templates (Appendix A), model selection rationale (Appendix B), and accessible data supports full reproducibility.

---

> ### Author Rebuttal · Authors · 2025-07-30
>
> Thank you for your thoughtful and positive review of our paper, “Thousand Voices of Trauma.” We are very encouraged by your support and your recognition of the paper's key strengths.
>
> We agree with your point that therapist evaluation across more transcripts would further strengthen the study. Due to the significant time and effort required for such detailed clinical annotations, we chose to focus on deep qualitative feedback from seven experts to guide our development. We then released the full dataset to the community for broader review and to enable future work, such as larger-scale evaluations.
>
> Thank you again for your positive assessment and helpful feedback.

---

> > ### Comment · Reviewer_M2aq · 2025-08-04
> > **Clarification Appreciated; Score Maintained**
> >
> > Thank you for the thoughtful and appreciative response. I understand the practical constraints surrounding the scaling of therapist evaluations and appreciate the authors’ rationale for focusing on in-depth qualitative feedback from a smaller expert sample.
> >
> > While I still believe that a more extensive clinical and cultural evaluation would enhance the dataset’s utility, the authors’ response is reasonable given the scope of this initial work. My concerns remain worth noting for future iterations, but they do not significantly impact the overall contribution or quality of the paper.
> >
> > I am keeping my score unchanged.

---

### Official Review · Reviewer_KuVq · 2025-06-30

**Rating:** 5
**Confidence:** 2

**Summary:**

The submission introduces a large-scale synthetic dataset of 3,000 therapy conversations based on Prolonged Exposure therapy for PTSD. The dataset includes diverse demographics, 20 trauma types, and 10 trauma-related behaviors, generated using Claude Sonnet 3.5. Key contributions include:

- Scale and Diversity: First large-scale PE therapy dataset with balanced representation across age, gender, and ethnicity.
- Clinical Depth: Structured into six PE therapy phases, validated by experts for fidelity.
- Benchmark Framework: Includes metrics (Pearson, DTW, RMSE) to evaluate AI models' ability to track emotional trajectories.
- Privacy-Preserving: Synthetic data avoids ethical/legal barriers of real patient data.

The dataset aims to support AI development for mental health tools and clinician training. Limitations include predefined categories and variability in synthetic generation.

**Dataset Code Accessibility:**

Yes

**Dataset Code Comments:**

Motivation & Evidence:

- The dataset (Thousand Voices of Trauma) is publicly accessible (URL provided in Sec. 7)
- Benchmark reproducibility:
  -  Metrics (Pearson/DTW/RMSE) and evaluation protocol are fully defined (Sec. 5).
  -  Code for scoring and baseline comparisons is included (Sec. 7).

I do not find major access/reproducibility barriers.

**Ethical Comments:**

- Data quality and representativeness: While diverse, synthetic data may not fully capture real-world therapeutic nuances (Sec 3.2, Fig 5).
- Discrimination, bias, and fairness: Fixed demographic distributions (Sec 3.2) risk overlooking intersectional biases.
- Safety and security: Potential misuse of trauma-related synthetic data in non-clinical settings (Sec 8).

The paper addresses these proactively (e.g., expert validation, usage guidelines) but flags them as inherent synthetic-data challenges. No evidence of other concerns (human subjects, deception, etc.).

**Ethical Considerations:**

Yes, there are ethics concerns that require attention by the authors

**Ethics Flags:**

["Data quality and representativeness", "Safety and security", "Discrimination, bias, and fairness"]

**Final Justification:**

The authors successfully resolve all my concerns, and I also refer to the discussion between authors with other reviewers. Thus, I decide to improve my rating to 5. Nonetheless, since I'm not very familiar with medical background, I can only set my confidence as 2.

**Limitations Weaknesses:**

- **Limited real-world validation**: While expert-rated (Sec 4, Fig 5), synthetic dialogues lack direct comparison to real PE therapy transcripts. Suggestion: Include side-by-side fidelity analysis with de-identified clinical data in future work.
- **Fixed demographic distributions**: Synthetic profiles follow predefined age/gender probabilities (Sec 3.2, Fig 1), potentially missing nuanced cultural or intersectional factors. Suggestion: Incorporate adaptive sampling for underrepresented groups.
- **Benchmark dependency on synthetic baselines**: Emotional trajectory metrics rely on Claude Sonnet’s scores (Sec 5, Table 1), not human clinician judgments. Suggestion: Add human-rated benchmarks to validate scoring alignment.

**Strengths Contributions:**

- **First large-scale synthetic PE therapy dataset** (3k dialogues) with diversity (20 trauma types, 10 behaviors, balanced demographics).
- **Clinically validated**: Structured into 6 PE phases, expert-rated for realism (Fig 5).
- **Novelty**: Metrics (Pearson/DTW/RMSE) to evaluate AI emotional tracking (Table 1).
- **Privacy**: Synthetic data avoids real-patient ethical issues (Sec 8).
- **Presentation**: Logical flow, informative figures (Figs 1-3,6), limitations discussed (Sec 9).

---

> ### Author Rebuttal · Authors · 2025-07-30
>
> Thank you for your thoughtful feedback and constructive suggestions. We are grateful for your positive assessment and are encouraged by your recognition of the paper's strengths, including our transparency on ethical flags.
>
> In response to your suggestion for a real-world fidelity comparison, we would like to highlight our companion paper. As noted in the main paper (see line 206 and 207), we have conducted such a validation in our companion study, *"How Real Are Synthetic Therapy Conversations?"* (B N et al., 2025). This work directly compares our synthetic dialogues with real PE transcripts and demonstrates strong fidelity in structure, flow, and emotional dynamics. The companion paper has undergone peer-review at another venue and is due for acceptance. We will emphasize this more prominently in the final version.
>
> Regarding the benchmark, we agree that a human-rated benchmark is a compelling future direction. While Claude Sonnet scores power the benchmark for scalability, we do validate these results via alignment with expected PE patterns (e.g., emotional peaks during exposure).
>
> Thank you again for your valuable review.

---

> > ### Comment · Reviewer_KuVq · 2025-08-05
> >
> > I have read the rebuttal, and I want to thank the reviewer for their feedback. In addition, the mentioned companion study is indeed relevant and valuable.

---

### Official Review · Reviewer_J7Rs · 2025-07-02

**Rating:** 4
**Confidence:** 4

**Summary:**

This paper presents a synthetic dataset of 3,000 Prolonged Exposure (PE) therapy conversations for PTSD treatment, generated using Claude Sonnet 3.5. The dataset comprises 500 unique patient profiles, each with six conversational phases following PE therapy protocols, and includes diverse demographics and trauma types. The authors also propose an emotional trajectory benchmark for evaluating AI models' ability to track emotional intensity across therapy sessions.

**Dataset Code Accessibility:**

Yes

**Dataset Code Comments:**

The code and data are accessible.

**Ethical Considerations:**

No, there are no or only very minor ethics concerns

**Final Justification:**

The authors' rebuttal helps address my primary concern about generation quality. However, the evaluation framework remains insufficiently comprehensive for the complexity of therapy conversations, and the low scores on certain aspects from clinical experts and limited diversity coverage of patient/therapist types (as raised by other reviewers as well) continue to raise questions about the dataset's readiness for downstream applications in sensitive mental health contexts. But I do believe the work represents a valuable contribution to privacy-preserving mental health research and addresses an important data scarcity problem, and the dataset is a reasonable first step that would benefit from more rigorous quality assurance before widespread adoption in clinical AI development.

**Limitations Weaknesses:**

1. The work shows limited novelty. The core contribution appears to be straightforward prompt engineering with Claude Sonnet 3.5. The prompt templates shown in Appendix A are relatively simple and lack sophisticated design considerations. For instance, the prompts rely on basic instructions like "Generate a detailed creative dialogue" without incorporating nuanced therapeutic principles or conversation dynamics.
2. The evaluation of conversation quality is overly simplistic. While the authors conduct expert evaluation with 7 therapists (Section 4), they only assess four high-level dimensions. The evaluation lacks critical aspects such as (I'm not familiar with the certain clinical setting, but I believe there should be a lot more):
    - Adherence to specific PE protocol steps
    - Clinical accuracy of therapist interventions
    - Appropriateness of emotional escalation/de-escalation
    - Linguistic naturalness and conversation flow

3. Expert Evaluation Results are concerning. The "Perceived Value" ratings are particularly troubling, with experts rating the conversations mostly being "Not valuable" and "Somewhat valuable". This raises serious questions about the dataset's utility for downstream applications.
4. While the abstract mentions that "Clinical experts... suggested refinements for greater authenticity," there's no evidence that the authors incorporated this feedback to improve the generation process. The study appears incomplete without iterative refinement based on expert input.
5. Given the quality concerns about the synthetic conversations, the emotional trajectory benchmark (Section 5) becomes problematic. It's unclear whether models are learning to track genuine therapeutic emotional progression or artifacts of synthetic generation.

**Strengths Contributions:**

- The dataset provides a large-scale resource (3,000 conversations) with well-defined structure following established PE therapy protocols (Foa et al., 2007), addressing the scarcity of mental health conversation data noted in prior work.
- The dataset demonstrates thoughtful consideration of demographic representation across age, gender, and ethnicity, which is important for developing inclusive AI systems.
- By using synthetic data generation, the work circumvents significant ethical and legal barriers associated with real patient data, enabling broader research access while maintaining privacy.
- The authors consulted with licensed psychotherapists to design prompts based on PE therapy protocols, providing some degree of clinical validity.

---

> ### Author Rebuttal · Authors · 2025-07-30
>
> Thank you for your time and detailed feedback on our paper, "Thousand Voices of Trauma." We are encouraged that the reviewers recognized the paper’s key strengths, and we appreciate the opportunity to clarify a few key points that we believe address your core concerns.
>
> ### 1. Prompt Engineering is Structured and Clinically Informed
> We would like to clarify that our prompts are not simple or generic. While the appendix overview begins with a general instruction, each prompt is shaped by multiple therapy-specific constraints. For example, *A.5: Full Imaginal Exposure (starting line 423)* instructs the model to:
>
> * Simulate the core and most emotionally intense component of Prolonged Exposure therapy (Full Imaginal Exposure)
>
> * Vary the client's emotional expressions (e.g., nervousness, determination, hesitation) across sessions to avoid repetitive patterns
>
> * Incorporate SUDS (Subjective Units of Distress Scale) monitoring and grounding techniques to help the therapist track and manage distress
>
> * Maintain narration in the present tense for greater emotional vividness
>
> * Include moments where the client struggles emotionally or physically, and ensure the therapist responds with empathy and encouragement
>
> * Therapist provides supportive interventions when necessary
>
> * Sustain a realistic duration of 30–45 minutes, without mid-session stopping or artificial truncation
>
> * Reflect a naturalistic conversational flow with pauses, filler words, and dynamic pacing
>
> * Enforce a clear turn-taking structure in which the client speaks at length and the therapist responds concisely
>
> We believe these instructions reflect core PE therapy best practices and go beyond "straightforward" prompting.
>
> ### 2. Real-World Validation Through Companion Paper
> We appreciate the concern regarding real-world comparison. As noted in the paper (see line 206 and 207), we have conducted such a validation in our companion study, *"How Real Are Synthetic Therapy Conversations?"* (B N et al., 2025). This work directly compares our synthetic dialogues with real PE transcripts and demonstrates strong fidelity in structure, flow, and emotional dynamics. The companion paper has undergone peer-review at another venue and is due for acceptance. These results support the clinical validity of our dataset, and we will emphasize this more prominently in the final version.
>
>
> ### 3. Context for Expert Evaluation
>
> We agree the "Perceived Value" scores could appear concerning without context. However, this evaluation was formative, used early in development to guide iterative improvements (e.g., reducing generic therapist responses, enhancing conversational realism). Other metrics were consistently strong: Experts rated Content Depth as "Detailed/Very Detailed" and Patient Engagement as "Moderate to Extreme," reflecting positive perceptions of realism and narrative quality. We’ll clarify this context to better reflect the developmental nature of the evaluation.
>
> We hope these clarifications address your main concerns regarding prompt structure, real-world fidelity, and the context of the expert evaluation. We believe these additions reinforce the rigor and relevance of our work. Thank you again for your thoughtful review.

---

> > ### Comment · Reviewer_J7Rs · 2025-08-07
> >
> > I appreciated the clarification provided by the authors. The response is convincing. The companion study is helpful for providing more context on this work. I will update my rating to positive.

---

> > > ### Author Response · Authors · 2025-08-08
> > >
> > > Thank you for considering our clarifications. We appreciate your support.

---

### Comment · Area_Chair_twQo · 2025-08-01
**Please read the rebuttal and get engaged**

Dear reviewers,

The authors have submitted the rebuttal to your review. Now we are in the reviewer-author discussion phase. Please read the rebuttal and indicate if it has addressed your concerns. If so, please update your review comment accordingly; if not, please let authors know your further concerns. Your active participation is of critical importance to the conference. Please get engaged as soon as possible. Thanks!

Your AC

---

### Comment · Area_Chair_twQo · 2025-08-05

Dear reviewers,

As you may have received, this year's NeurIPS made a lot of changes to force you to read the rebuttal and keep discussion with the authors. Therefore, if you have not, please do that as soon as possible (so you will not receive reminders like this in the future). I will actively monitor the discussion phase until you do. Please engage. Your contribution is of immense importance to the reviewing period.

AC

---

### Note · Authors · 2025-08-12

Dear Area Chair and Reviewers,

Thank you for the time and effort you have invested in our submission. The rebuttal period has been valuable for clarifying our goals and reflecting on your thoughtful feedback, which has greatly strengthened the paper.

We noticed that for three reviewers, the initial score is no longer visible, indicating updates after the discussion. For Reviewer KuVq’s review, the initial rating of 4: Borderline Accept is still displayed. Given your encouraging comments - that our rebuttal was helpful and that the companion study is “indeed relevant and valuable”, we wanted to politely confirm whether this score reflects your final evaluation.

Thank you again for your constructive input and guidance.

With appreciation,

Authors

---

### Decision · Program_Chairs · 2025-09-18

**Decision:**

Accept (spotlight)

**Comment:**

This paper introduces Thousand Voices of Trauma, a large-scale synthetic dataset of 3,000 Prolonged Exposure (PE) therapy conversations for PTSD, generated using Claude Sonnet 3.5. Reviewers highlighted the dataset’s methodological grounding in PE protocols, diversity of trauma types and demographics, and structured conversational phases that reflect clinical fidelity (Reviewer KuVq; Reviewer M2aq). The work stands out for addressing the scarcity of accessible therapy data by creating a privacy-preserving synthetic resource (Reviewer J7Rs; Reviewer y3Hw). Clinical experts were involved in both the prompt design and the evaluation process, enhancing the dataset’s credibility. The inclusion of an emotional trajectory benchmark with well-defined metrics (Pearson, DTW, RMSE) was praised as a novel contribution that enables systematic assessment of AI systems’ capacity to track emotional intensity (Reviewer KuVq; Reviewer y3Hw). Overall, reviewers agreed that this paper provides a valuable, reproducible, and ethically responsible contribution that has the potential to support AI applications in mental health, clinician training, and evaluation research.

At the same time, reviewers pointed out several limitations that temper the impact. Reviewer J7Rs emphasized that the generation process relies on relatively simple prompt engineering and that expert ratings revealed concerns about perceived value, raising questions about the dataset’s readiness for downstream clinical applications. Both Reviewer KuVq and Reviewer M2aq noted that the validation is limited in scope: only seven therapists evaluated a small subset of transcripts, and there was no direct comparison to real PE therapy transcripts. Reviewer M2aq further raised concerns about cultural nuance, as the random assignment of ethnicity may fail to capture intersectional or culturally specific trauma expressions. Reviewer y3Hw added that details about model settings in the benchmark (e.g., temperature) were unclear, potentially affecting results. Despite these shortcomings, the consensus is that the strengths outweigh the weaknesses, especially as a first release. The work is technically solid, thoughtfully presented, and represents an important step toward privacy-preserving, clinically informed AI datasets.